# Efficacy of artemisinin-based and quinine-based treatments for uncomplicated falciparum malaria in pregnancy: a protocol for systematic review and individual patient data (IPD) meta-analysis

Makoto Saito,[1,2] Rashid Mansoor,[1,2] Kalynn Kennon,[1,2] Rose McGready,[2,3] François Nosten,[2,3] Philippe J Guérin,[1,2] Kasia Stepniewska[1,2]

¹WorldWide Antimalarial Resistance Network (WWARN), Oxford, UK
²Centre for Tropical Disease and Global Health, Nuffield Department of Medicine, University of Oxford, Oxford, UK
³Shoklo Malaria Research Unit, Mahidol-Oxford Tropical Medicine Research Unit, Faculty of Tropical Medicine, Mahidol University, Mae Sot, Thailand

**Correspondence to**
Dr Makoto Saito;
makoto.saito@wwarn.org

## ABSTRACT

**Introduction** Pregnant women are more vulnerable to malaria leading to adverse impact on both mothers and fetuses. However, knowledge on the efficacy and safety of antimalarials in pregnancy is limited by the paucity of randomised control trials and the lack of standardised protocols in this special subpopulation. Pooling individual patient data (IPD) for meta-analysis could address in part these limitations to summarise accurately the currently available evidence on treatment efficacy and risk factors for treatment failure.

**Methods and analysis** To assess the treatment efficacy of artemisinin-based and quinine-based treatments for uncomplicated falciparum malaria in pregnancy, seven databases (Medline, Embase, Global Health, Cochrane Library, Scopus, Web of Science and Literatura Latino Americana em Ciências da Saúde) and two clinical trial registries (International Clinical Trials Registry Platform and ClinicalTrial.gov) were searched. Both interventional and observational cohort studies following up for at least 28 days will be included. IPD of the identified eligible published or unpublished studies will be sought by inviting principal investigators. Raw IPD will be shared through the web-based secure platform developed by the WorldWide Antimalarial Resistance Network using the established methodology. The primary objective is to compare the risk of PCR-corrected treatment failure among different treatments and to find the risk factors. One-stage IPD meta-analysis by Cox model with shared frailty will be conducted. A risk of bias assessment will be conducted to address the impact of unshared potential data and of the quality of individual studies. Potential limitations include difficulty in acquiring the IPD and heterogeneity of the study designs due to the lack of standard.

**Ethics and dissemination** This IPD meta-analysis consists of secondary analyses of existing anonymous data and meets the criteria for waiver of ethics review by the Oxford Tropical Research Ethics Committee. The results of this IPD meta-analysis will be disseminated through open-access publications at peer-reviewed journals. The study results will lead to a better understanding of malaria treatment in pregnancy, which can be used for clinical decision-making and conducting further studies.

**PROSPERO registration number** CRD42018104013.

---

### Strengths and limitations of this study

► This study will be the first individual patient data (IPD) meta-analysis on the efficacy of currently recommended antimalarials in pregnancy incorporating IPD from both randomised control trials (RCTs) and single-arm cohort studies, overcoming the limitation of aggregated data meta-analysis that can only include RCTs.

► IPD that are standardised in the same format and analysed in a uniform way with adjustment of covariates will, in contrast to aggregated data, allow us to compare the efficacy of different treatments as well as to find risk factors for treatment failure in this vulnerable but understudied population.

► Limitations of this IPD meta-analysis include the potential difficulty in acquiring the IPD and the heterogeneity of the study designs, study population and parasite population. A risk of bias assessment will be conducted to address the impact of unshared potential data and of the quality of individual studies.

---

## INTRODUCTION

About 60% of all pregnancies are estimated to take place in malaria-endemic areas.[1] In addition, pregnant women are among the most vulnerable groups for malaria infection leading to higher morbidity and mortality of both mothers and fetuses.[2] Although around 1500 studies on the efficacy of antimalarials in malaria treatment have been conducted,[3] pregnant women have been excluded from the majority of clinical trials in the past, mainly because of safety concerns for the fetus.

Due to the lack of evidence for both efficacy and safety of antimalarials in pregnancy, quinine (with clindamycin if available), rather than artemisinin-based combination therapy (ACT), has been recommended as the first-line treatment of uncomplicated *Plasmodium falciparum* malaria for pregnant women in the first trimester by WHO.[4] However, recent studies measuring the safety of artemisinin derivatives during pregnancy, including in the first trimester, have shown reassuring results,[5–8] and it is likely that ACT will be recommended as the first-line treatment option for pregnant women regardless of the trimester in the next WHO treatment guidelines.[9] Evidence on the treatment efficacy during pregnancy needs to be assembled.

The efficacy and safety of antimalarials in pregnancy can be different from the results from the non-pregnant populations because of altered immunity, physiological change in pharmacokinetics and sequestration of parasites to the placenta. The risk factors for treatment failure in pregnancy need to be assessed to improve clinical care in pregnancy. However, there are no agreed guidelines on how to assess the efficacy in pregnancy while it is standardised in the non-pregnant patients by WHO.[10] This lack of standard methodology makes it challenging to conduct efficacy studies in pregnancy and leads to the variability of assessing and reporting the outcomes.[11 12] Taken together, the current situation limits conducting aggregated data meta-analyses.[12]

The WorldWide Antimalarial Resistance Network (WWARN) has established a unique individual participant data (IPD)-sharing platform facilitating large-scale pooled meta-analyses. We plan to include both published and unpublished studies exploring the efficacy and safety of the treatment of malaria during pregnancy. We will conduct a one-stage IPD meta-analysis on the currently recommended antimalarial drugs, that is, artemisinin-based and quinine-based treatments, used for the treatment of uncomplicated falciparum malaria in pregnancy.

### Objectives

The aim of this study is to evaluate and compare treatment outcomes of artemisinin-based and quinine-based treatment for uncomplicated falciparum malaria in pregnancy.

Primary objectives are

► To compare antimalarial efficacies among artemisinin-based and quinine-based treatments.
► To identify risk factors associated with treatment failure.

Secondary objectives are

► To assess the relationship between the dosing (dose per body weight) of artemisinin-based treatments and treatment efficacy.
► To evaluate the risk of gametocyte carriage following artemisinin-based and quinine-based treatments.
► To evaluate the safety and tolerability of artemisinin-based and quinine-based treatments.

## METHODS AND ANALYSES
### Criteria for study eligibility
#### Types of studies

► Prospective clinical efficacy studies with a minimum 28-day active follow-up.
► Both interventional and observational cohort studies regardless of the number of treatment arms (ie, comparative or single arm).
► Genotyping conducted for distinguishing recrudescence and reinfection.

The following studies will be excluded.

► ≤10 eligible pregnant women.
► Conducted in non-endemic countries (ie, returned travellers).

#### Types of participants

► Pregnant women in any trimester.
► Parasitologically confirmed *P. falciparum* parasitaemia.
► Either asymptomatic or symptomatic.

#### Types of intervention/exposure and controls

► Treated with artemisinin-based or quinine-based treatments.

#### Types of outcomes

► Parasitological and clinical efficacy.
► Adverse events.

### Information sources and search strategy

A systematic literature review was conducted to identify the potential studies to be included in this IPD meta-analysis. Seven databases (Medline, Embase, Global Health, Cochrane Library, Scopus, Web of Science and Literatura Latino Americana em Ciências da Saúde) and two clinical trial registries (International Clinical Trials Registry Platform and ClinicalTrial.gov) were used. Both published and unpublished grey literature such as conference abstracts and registered trials were included. This systematic review and IPD meta-analysis is registered to PROSPERO, and the search terms and conditions are available there.

Briefly, the search combined five components: malaria; pregnancy; treatment or names of antimalarial drugs; study design (interventional or observational cohort studies) and outcome types (efficacy) without limitation on publication year or language. The result of the literature search was published elsewhere.[12] The initial search was conducted on 9 July 2016. The final search will be updated in April 2019.

### Data acquisition and data management
#### Collecting IPD

Principal investigators of the published and unpublished studies identified by the systematic literature review will be invited to share their IPD with WWARN. Emails will be sent to the corresponding authors on at least three occasions asking whether they are willing to join the study group. A secure web-based platform has been developed by WWARN, and IPD will be uploaded after agreeing to

the terms and conditions of the submission, retaining the ownership and full control of their shared data.[13] Data are fully anonymised and handled in compliance with the UK Data Protection Act to protect personal information and patient privacy. Original data are stored on a secure server hosted by the University of Oxford.

### Data management

Raw data will be curated in a standardised format using the WWARN Clinical Module Data Management Plan to facilitate pooled IPD meta-analyses.[14] After checking the raw data, any queries on the availability of data, ambiguity of the variables or potential errors will be resolved by asking the data contributors. The protocol of the original studies will be sought from the data contributors or the publication when available. The standardised dataset will be used for the analyses.

### Statistical analysis plan

#### Study populations

Pregnant women will be eligible for the purpose of this analysis if the following information is available:

► Confirmed pregnancy status on day 0 of the treatment.
► Type, date and dose of antimalarial drugs (artemisinin-based or quinine-based treatments).
► Patient age and estimated gestational age (or trimester of pregnancy) on day 0.
► Date of the last day of follow-up or length of follow-up.

The following patients will be excluded:

► No or missing data on parasitological confirmation of *P. falciparum* infection at enrolment.
► Presenting with severe malaria symptoms at enrolment as defined by WHO,[4] except hyperparasitaemia and severe anaemia, which will be included.

#### Outcomes

The primary outcome will be the PCR-corrected *P. falciparum* treatment failure. Secondary outcomes will include any recurrence of malaria (PCR-uncorrected treatment failure); parasite clearance; gametocyte carriage during follow-up and adverse events that developed after drug administration. Pregnancy outcomes and placental malaria may be assessed if enough data are gathered.

Recurrences of *P. falciparum* will be distinguished by PCR into recrudescence (treatment failure) and reinfection.[15] Indeterminate PCR will be excluded, and reinfection will be regarded as being censored on the day of recurrence in survival analyses for PCR-corrected outcomes following the WHO guidelines.[10] In studies where peripheral malaria smears were examined regularly (eg, every week), the time of parasite recurrence will be defined as the time of the first positive parasite smear after the parasite clearance following the treatment. For pregnant women with no recurrent parasitaemia recorded, the day of their last negative smear will be regarded as their last visit and censoring time. In the case of intermittent follow-up (eg, missed follow-ups), the following rules will be applied:

1. Blood smears will be assumed negative between the two negative observations.
2. If a patient came back to be followed up with a positive smear, the date of positive parasitaemia will be assumed to be the date of observation if this date is within 28 (±3) days from the last observation.
3. If parasite clearance is not recorded after treatment but the positive parasite count is recorded at least 7 days after starting the treatment, the day of the first positive count will be regarded as the day of recurrence.

Definitions of status and other censorship are detailed in the Clinical Module Data Management Plan[14] except for the above modification. The presence of parasitaemia within the first 7 days will not be regarded as treatment failure for quinine-based treatment because quinine is given for 7 days.

Adverse symptoms will include abdominal pain, dizziness, headache, body pain/myalgia, weakness/fatigue, vomiting, nausea, anorexia and tinnitus if data permit.

### Variables and their definitions

The following baseline characteristics of patients will be included as appropriate if enough data are shared: age; estimated gestational age (or trimester); parity or gravidity; weight (weight before pregnancy and weight at treatment); body mass index; baseline parasitaemia; presence of fever (body temperature >37.5°C); haemoglobin (or haematocrit); anaemia (haemoglobin <110 g/L or haematocrit <30% for anaemia and haemoglobin <70 g/L or haematocrit <20% for severe anaemia)[16]; gametocytes on presentation; history of malaria or antimalarial use; description of infection (mixed species infections); total mg/kg dose for each drug component; and supervision of drug administration. The doses of drugs received will be calculated from the number of tablets administered to each patient. If the actual number of tablets received was not recorded, doses according to the protocol will be used. Only those who completed the standard dose will be included in the primary analysis. The proportion of partial treatment will be presented.

For each study, study locations and local transmission intensity will be considered. The study sites will be classified into three categories: low, medium and high malaria transmission based on the parasite prevalence estimates obtained from the Malaria Atlas Project for specific location and year of study.[17 18]

*Plasmodium vivax* intercalated infection (ie, *P. vivax* monoinfection before the recurrence of *P. falciparum* parasitaemia) will be regarded as censored if the original study did not test PCR for falciparum recurrences after intercalated vivax infection, following the WHO guidelines.[10] If the original study tested PCR for falciparum recurrences regardless of intercalated vivax infection, vivax infection will be regarded as a time-dependent covariate.

### Descriptive summaries

A summary of the studies and baseline characteristics of the patients included in the analysis will be presented. The number of available patients will be summarised for all variables listed above, proportion will be used for categorical or binary variables and mean and SD (or median and IQR) will be used for continuous variables.

PCR-corrected and uncorrected outcomes will be used to compute the Kaplan-Meier (K-M) estimates for each study site. The efficacy of each treatment will then be summarised at fixed time points (ie, on day 28, 42 and 63) by the aggregated meta-analysis approach.

### Analysis of primary outcome

A one-stage IPD meta-analysis using the Cox model with shared frailty for study sites will be conducted to identify the risk factors for treatment failure as well as comparing different treatments. For repeated episodes, if any, multilevel mixed-effects model (if there are enough data) or the previous history of malaria will be used. If data permit, a non-linear relationship will be examined for continuous variables.[19] Cox-Snell and Schoenfeld residuals will be examined to determine the appropriateness of model fit and proportional hazard assumption, respectively. Alternative statistical approaches such as flexible parametric models or introducing an interaction term with time will be considered if the proportionality assumption is not satisfied.

### Analyses of secondary outcomes

Analysis of secondary outcomes will be carried out provided enough data are present; else, only summary statistics will be reported. Analyses similar to the primary outcome will be conducted for PCR-uncorrected treatment failure (ie, any recurrence of malaria).

Parasite clearance will be assessed as the proportions of patients cleared asexual falciparum parasitaemia on day 1, 2 and 3. Univariable and multivariable mixed-effects logistic regression models (or Cox models for the time to parasite clearance) will be used to identify the risk factors associated with parasite positivity status.

Gametocyte carriage will be assessed as the proportion of patients with *P. falciparum* gametocytes on day 0, 3, 7, 14, 21 or 28. Proportions after day 0 will be stratified by the presence of gametocytes at baseline. If enough data are available, mixed-effects logistic regression models will be used to assess the risk factors for gametocytes carriage after treatment stratified by the presence of gametocytes at baseline.

Adverse effects will be assessed as the proportion of patients who developed symptoms after the treatment initiation. Proportions of patients who developed symptoms after day 0 will be stratified by whether or not that symptom was present before the treatment initiation. If enough data are available, mixed-effects logistic regression models will be used to assess the risk factors for adverse symptoms developed after the treatment initiation. Symptoms on day 0 (before treatment) will be added as a covariate. Primarily, the symptoms developed in the first week will be included.

### Variable selection

For any regression models, the following strategy recommended by Collet[20] will be used to determine independent risk factors. Initially, all possible risk factors will be examined in the univariable model to assess if any of the variables are related to the treatment outcome. All significant variables with a p value of ≤0.05 will then be added to the baseline model. The variables with a p value of >0.05 will be excluded from the baseline model one by one starting from the variable with the largest p value. Once only significant covariates will remain in the model, all excluded variables will be added to this model one by one to check whether there will be any variables that become significant in the presence of other risk factors. Likelihood ratio test and Akaike's information criterion will be used to compare nested and non-nested models, respectively. Treatment and baseline parasitaemia will be included in the multivariable models on treatment efficacy as a priori forced variables regardless of the statistical significance. Variables that are missing more than 50% will not be included in multivariable analyses.[21] Interaction between gravidity (parity) and endemicity, or age and endemicity will be assessed if age or gravidity is included in the multivariable model, as the impact of age and gravidity (ie, pregnancy-specific immunity) can be different depending on the endemicity.[22]

### Assessment of statistical heterogeneity across studies

The multilevel logistic or Cox models would be used for explaining the study-site heterogeneity. Heterogeneity across study sites will be statistically assessed as the variance of the shared frailty term estimated in the Cox model or variance of the random intercepts in logistic regression. Additionally, the intraclass correlation in logistic regression model will be reported.

### Subgroup analyses

Analyses will be conducted by malaria transmission intensity and by treatment (for assessing dose impact of each drug) if data permit.

### Sensitivity analyses

Two types of sensitivity analyses will be performed. First, a model will be refitted with excluding one study at a time to identify any influential studies. Second, to assess the impact of covariates with missing values, multiple imputation may be used.[21]

### Strength of the body of evidence/risk of bias across studies

The risk of bias within and across the included studies will be assessed following the GRADE guidelines.[23] Publication bias will be evaluated by a funnel plot of the log-transformed hazards ratio (OR or proportion)[24] if more than 10 studies are included.[25] Despite the effort, all the studies identified in the systematic review may not be shared and included in this IPD meta-analysis. The bias by the studies

that are unable to be included in the analyses will be evaluated.[26] The reported aggregated efficacy will be extracted from the publication and compared with the studies included. A two-stage meta-analysis combining shared and unshared data will be attempted if data permit.[27] The impact of artemisinin resistance in the study year at the study site will be evaluated by using the reported prevalence on molecular resistance marker (K-13).

### Further development of statistical analysis plan

The main analysis is planned as described above. Modification or additional analyses may be required as the data collection progresses. Updated statistical analysis plans will be available at the WWARN website if an amendment is required.[28]

### Software

Statistical analysis will be conducted using R (The R Foundation for Statistical Computing, Vienna, Austria) or Stata MP V.15.1 (StataCorp).

### Patient and public involvement

This IPD meta-analysis will use existing secondary data. Patients and public were not involved in the design, recruitment or conduct of this IPD meta-analysis. The results of this study will be shared with the primary investigators of the shared studies and disseminated as publications in open-access journals.

### Dissemination

Findings will be reported following the Preferred Reporting Items for Systematic Reviews and Meta-Analyses (PRISMA)-IPD statement[29] at peer-reviewed journals with open access. The progress will be updated on our study group website.[28] This protocol is reported following PRISMA Protocols statement.[30 31] Any publications based on the findings of this IPD meta-analysis will be in accordance with the guidelines of the International Committee of Medical Journal Editors.

### DISCUSSION

This IPD meta-analysis will update the previous aggregated data meta-analyses that included only four or five randomised control trials.[12 32] In IPD meta-analyses, data from single-arm interventional or observational cohort studies can be included. As the data can be standardised and analysed in a uniform way, IPD meta-analyses are particularly useful when there is no standard study design such as in this case. Risk factors associated with treatment failures particularly the dosing of the currently used treatments can be assessed in IPD meta-analyses but rarely in aggregated data meta-analyses. Although meta-analyses of secondary data cannot include variables that were not assessed in the original studies, the results of this IPD meta-analysis can identify the pregnant women in need of close clinical monitoring based on what is commonly assessed. Despite the increased time and effort of gathering and standardising the IPD, the advantages of IPD meta-analysis outweigh particularly for answering research questions on these neglected or understudied populations.

WWARN has developed the secure and equitable data-sharing platform and the international collaborative network of malaria researchers worldwide over the last decade. With this unique collaborative effort, we hope that these findings will lead to the improvement of clinical management of this vulnerable but understudied population.

**Acknowledgements** We would like to thank Dr Mary Ellen Gilder for her contribution to the previous systematic literature review and Dr Prabin Dahal for his comments on the draft.

**Contributors** MS, RMc and PJG conceived the idea. MS and RM drafted the manuscript. KK, FN, RMc, PJG and KS critically revised the manuscript. All authors have read and approved the final manuscript.

**Funding** The WorldWide Antimalarial Resistance Network is funded by the Bill and Melinda Gates Foundation and the ExxonMobil Foundation. SMRU is part of the Mahidol Oxford University Research Unit supported by the Wellcome Trust of Great Britain. MS is supported by the University of Oxford Clarendon Fund.

**Competing interests** None declared.

**Patient consent for publication** Not required.

**Ethics approval** This individual patient data meta-analysis met the criteria for waiver of ethics review as defined by the Oxford Tropical Research Ethics Committee (OxTREC) since the research consists of secondary analyses of existing anonymous data. Each study included in the analysis will have received local ethics approvals.

**Provenance and peer review** Not commissioned; externally peer reviewed.

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
