## [Reviewer comments · BMJ Open]

ARTICLE DETAILS

TITLE (PROVISIONAL)	The efficacy of artemisinin-based and quinine-based treatments for uncomplicated falciparum malaria in pregnancy: a protocol for systematic review and individual patient data (IPD) meta-analysis
AUTHORS	Saito, Makoto; Mansoor, Rashid; Kennon, Kalynn; McGready, Rose; Nosten, François; Guerin, Philippe; Stepniewska, Kasia

VERSION 1 - REVIEW

REVIEWER	AM van Eijk Liverpool School of Tropical Medicine, Liverpool, UK
REVIEW RETURNED	19-Nov-2018

GENERAL COMMENTS	The efficacy of artemisinin-based and quinine-based treatments for uncomplicated falciparum malaria in pregnancy: a protocol for systematic review and individual patient data (IPD) meta-analysis, by Makoto Satio et al. Overall This is a detailed protocol of an important individual participant data analysis for malaria treatment among pregnant women. The study is relevant and examines drugs that are currently used; it uses institutionalized mechanisms to make now datasets more widely available. The paper is well written. I only have some minor questions or requests for clarification. Abstract and article summary In the abstract you mention “the lack of standardised protocols in this sub-population”. In the article summary you also mention: “Limitations of this IPD meta-analysis include the potential difficulty in acquiring the IPD and the heterogeneity of the study designs, study population and parasite population. A risk of bias analysis will be conducted to address the impact of potential unshared data.” I am not sure what the guidelines are for both these sections but would appreciate if some limitations could also be part of the abstract. Introduction “..around 1500 studies..” How was this number identified? Line 84: This “Besides,” can be removed, because the assessment of risk factor is one of the primary outcomes, and not a “Besides”. Methods
---

	How are observational cohort studies different from single arm studies? If all participants receive an intervention, I thought single arm studies were still interventional? Just wondering about the word observational here. “P. vivax intercalated infection will be regarded as censored if the original study did not test PCR for falciparum recurrences after intercalated vivax infection. If the original study tested PCR for falciparum recurrences regardless of intercalated vivax infection, vivax infection will be regarded as a time-dependent covariate.” This section is not clear to me. “after intercalated vivax infection” does this mean, after the P. vivax infection was detected or at the same time that the P. vivax infection was detected? If the participant received treatment for P. vivax, this may affect the treatment results for P. falciparum, so I am not sure how you can use P. vivax as a time-dependent co-variate. Would you also add treatment for P. vivax as a covariate? “Likelihood ratio test (LRT) and Akaike’s Information Criterion (AIC) will be used to compare nested and non-nested models, respectively.” My apologies for possible lack in statistical background, but I would appreciate if it could be explained what the nested models are in this context, given that these are all treatment studies. “Interaction between gravidity (parity) and endemicity will be assessed, as the impact of gravidity (i.e. pregnancy-specific immunity) can be different depending on the endemicity.” Will age also be explored? In a previous study in an area of high malaria endemicity, we noted that low gravidity number was more important as a risk factor to get malaria, whereas young age was more important as a risk factor for treatment failure. Of course, there may be co-linearity between these two, which needs to be evaluated, but underlying mechanisms may be different, dependent on endemicity in the area. Will there be an attempt to match the included studies with available molecular resistance data to artemisinins or the partner drug at the time of and in the region of the study? This may perhaps only be relevant to the Thai-Burmese border. Overall: The variation between past and future tense is confusing throughout the manuscript, and could be more consistent. E.g. in variable selection: “Once only significant covariates remained in the model, all excluded variables will be added to this model one by one to check whether there are any variables that become significant in the presence of other risk factors.” The search-section is written in the past as if this has already been conducted. So if this is part of an ongoing study, the number of potential studies identified could perhaps be added.
--	---

REVIEWER	Matthew Ippolito Johns Hopkins University, USA
REVIEW RETURNED	24-Nov-2018

GENERAL COMMENTS	REVIEWER COMMENTS
-------------------

The efficacy of artemisinin-based and quinine-based treatments for uncomplicated falciparum malaria in pregnancy: a protocol for systematic review and individual patient data (IPD) meta-analysis

BMJOPEN-2018-027503

TITLE AND GENERAL COMMENTS

This is a protocol for an important and very timely review of antimalarial drugs for malaria-in-pregnancy, carried out by leading experts in the field and coming on the heels of updated WHO recommendations. The authors' approach of combining results of RCTs and observational cohort studies is

1. Efficacy suggests results of Phase II clinical trials (e.g. potency, efficacy). My personal preference would be to remove this from the title perhaps simply dropping "The efficacy of" from the title.
2. For the authors' definition of malaria-in-pregnancy, they do not distinguish between uncomplicated malaria and asymptomatic malaria but rather group these together. They do not give sufficient attention to placental malaria, also a distinctive clinical entity. I suspect these patients groups may be different enough from each other, pathophysiologically and parasitologically, to warrant separation of the group in analyses. It may also be worthwhile to mention intermittent presumptive treatment in pregnancy (IPTp).
3. Can the authors please include their rationale for including only studies of ACTs and quinine rather than any antimalarial agent.
4. The authors should discuss differences between ACT and quinine pharmacokinetics and pharmacodynamics, and think about how these differences should inform their inclusions and exclusions, case definitions, treatment of missing time points, etc. Pharmacodynamics differ (e.g. parasite clearance is slower with quinine) and some ACTs have partner drugs with extremely long elimination half-lives (mefloquine, piperazine).

ABSTRACT

Line 30: The preferred term is "special population" when discussing pregnant women in relation to drug PK/PD (Line 30 and other locations throughout the article).

Line 31-21: Please add safety and tolerability in pregnancy (in addition to treatment efficacy and risk factors of treatment failure).

Line 36: Please spell out LILACS and other acronyms at their first instance throughout the manuscript.

Line 64: Another limitation is heterogeneity of transmission intensity among different study sites, and perhaps non-standard or missing covariates.

Line 65: The authors' risk of bias analysis will not only evaluate the impact of potential unshared data but also critically assess quality of studies i.e. a critical assessment of each included study's methods.

INTRODUCTION

Line 74: Please add, "Due to the lack of evidence for both efficacy and safety **of artemisinin-based combination therapy (ACT) in pregnancy** ..."

Line 77:

Can the authors explicitly state, for readers' sake, that the statistical methods they propose are the preferred method for combining results of observational cohort data with data from randomized controlled trials if this is the case? In the introduction can the authors please add a paragraph explaining to readers why they have chosen an individual pooled data (IPD) meta-analysis, and briefly discuss the benefit of this approach over aggregated meta-analysis approaches that more readers are familiar with?

(There is language in the Discussion that the authors can move, or adapt to, the Introduction.)

Line 78: I do not think it is necessary to speculate regarding future WHO guidelines. The authors can highlight that their study will provide additional evidence for consideration. The reference (#8) is to the 2015 guidelines so I would omit this.

Line 80: "...the bulk of evidence is limited" The authors should omit this sentence, or reword to explain what evidence is lacking (safety, efficacy, dosing/PK, etc.).

Line 84: Replace "Besides" with "In addition" or similar connector.

Line 85-88: This section is not clear to me. Why are assessments of clinical efficacy in pregnant women necessarily different from clinical assessments in other populations? WHO defines adequate clinical and parasitological response, early treatment failure, late clinical failure, and late parasitological failure, etc. Is there really a lack of standard methodology? Perhaps if the authors discuss the delineation between uncomplicated malaria, asymptomatic malaria, and placental malaria it will resolve this sticking point.

OBJECTIVES

Line 107-109: Since they are important secondary outcomes, perhaps the authors should review the literature and discuss gametocyte carriage in pregnant women, and risk of vivax malaria following treatment of falciparum malaria, in their Introduction

Line 108: Suggest, changing "risk of adverse events..." to "safety and tolerability of artemisin- and quinine-based treatments"

METHODS AND ANALYSES

Line 113: The authors may want to consider including studies even with shorter follow-up times than 28 d because there may be safety and efficacy outcomes, gametocyte outcomes etc. reported in those studies.

Line 116: The authors should pre-specify which genotyping approaches they will accept (e.g. all three of mps1, mps2, glurp).

Line 118: Can the authors justify why they are excluding studies with ≤ 10 eligible pregnant women? Is this related to assumptions about baseline hazard/shared frailty approach? If this is an arbitrary number, can the authors remove this exclusion?

Line 119: What about displaced or migrant populations? These populations are commonly excluded because of differences in premunition, etc.

Line 122: Is there a cutoff for parasitemia (e.g. $>200,000$ or $>4\%$ meets definition of severe). What about mono vs. mixed Plasmodium infections?

Line 123: The authors must think more carefully about this. Asymptomatic malaria is distinct from uncomplicated malaria and strong justification is needed if the authors intend to combine these two patient groups. They may want to exclude asymptomatic malaria or stratify by uncomplicated vs. asymptomatic malaria.

Lien 126: The authors forgot to include gametocyte-related outcomes. They may want to add hemoglobin recovery, QT prolongation. They could expand adverse events to include pregnancy complications, adverse events that affect the infant or fetus. They may want to include pharmacokinetic and pharmacodynamic data if sufficient studies with these measures are identified.

Line 130-142: The search be conducted within 12 months of the intended publication date of the meta-analysis itself. Can the authors update all of their searches (Scopus, LILACS, etc.) since these were last done over 2 years ago in July 2016. Please add the date range for the search including beginning date i.e. was it all studies since inception of each database?

Line 142: May want to add, "... based on studies identified by the final updated search **for which patient-level data are made available**"

Line 149: Can the authors please very briefly state, in 1-2 sentences, what these terms and conditions are? It can help readers understand why a certain PI may or may not have agreed to contribute data.

Line 151: It is just a minor suggestion but the authors may also want to include that the WWARN includes a mechanism for contributors and other researchers to request access to the WWARN data.

Line 155: Minor typo, "REsolved"

Line 161: Add additional bullets here for completeness e.g. gravidity/parity, directly observed therapy or not, recent antimalarial drugs (should this be an exclusion criterion?) including whether IPTp was administered or not, geographic region including information about transmission intensity, mono vs. mixed Plasmodium infection, parasite density, hemoglobin concentration, type of malaria (uncomplicated, asymptomatic, placental—see comments above), parasite clearance.

Line 172: Why are the authors keeping those with hyperparasitemia? These cases should be clinically regarded as severe malaria, even if they were not treated as such in a particular study. The authors could consider a subgroup analysis, or sensitivity analysis excluding these hyperparasitemic patients, if the authors retain them in their dataset.

Line 174: How will the researchers define "incomplete dose" and should the authors consider including those patients? This could become an important source of bias e.g. incomplete doses might be due to adverse event/intolerability of the medication, disease progression requiring change in therapy, or other events that the authors would want to capture.

Line 176-179: I believe the primary outcome should be stated as PCR corrected clinical outcome (ACPR, ETF, LCF, LPF). Do other secondary outcomes include adverse events, hemoglobin recovery, pregnancy outcome, placental malaria, infant/fetus AEs, etc.? This appears to be the first mention of parasite clearance (Line 178).

Line 178: Why is uncorrected Pf recurrence included here? The authors stated that they are only including studies for which genotyping data are included for PCR correction of clinical outcomes (i.e. to distinguish reinfection vs. recrudescence).

Line 181: The authors should consider competing risk analysis rather than censoring to deal with reinfections +/- indeterminate PCRs. Censoring those with indeterminate PCRs is going to potentially introduce bias, and note of this should be made under limitations.

Line 188: I do not think it is safe to assume a missing weekly smear between two negative weekly smears is also negative unless the authors are performing repeated failure/event survival analysis in which case a participant can enter and exit the risk set at different time points. (They propose repeated episodes analysis later in the manuscript, Line 234, but could be explicitly mention in the Introduction or other, earlier section.) However, I understand their approach to define any LPF as a failure event; and since it is not safe to assume that a missing smear = negative even if couched between two negative smears, then those participants should probably be censored at the last available observation before the missing observation.

Line 189-191: Why 31 days? The authors should choose a range based on parasite biology and drug pharmacodynamics (e.g. some ACT partner drugs, and quinine, have very long half-lives).

Line 196: Correct "Early late treatment failure" to "Early treatment failure"

Line 198: The authors should capture all reported AEs rather than a list of prespecified AEs, or otherwise justify why they are using a proscribed list. It would also be interesting to discuss post-artemisinin hemolysis which can also occur with oral agents.

Line 206: What about gametocytes that emerge after start of treatment in those who were initially without gametocytemia?

Line 208: Are the authors capturing information on recent antimalarial treatment within the past week or 2 weeks, etc.?

Line 216: The authors may want to consider *P. vivax* infection as a competing risk, rather than censoring (see comment above).

Line 224: Range, or interquartile range? What about geometric mean for parasite counts (preferred by many in the field), or PK data?

Line 228: My instinct is to drop the analysis at fixed time points (28, 42, 63 d). Results can be misleading—the authors are going through the rigor of collecting longitudinal data so why reduce it to cross-sectional data even in an exploratory analysis? I would favor dropping this altogether.

Line 250: For mixed effects, can the authors state which are fixed and which are random?

Line 254: Adverse events, not adverse symptoms.

Line 259: "Symptoms developed at any time during the study period may be added." Please change to "Signs or symptoms", and please explain when they will or will not be added, or omit this sentence altogether.

Line 262: This approach is falling out of favor (referring to the authors' proposed strategy for selecting covariates). The authors refer to "risk factors" (predictors) but ignore potential confounders, mediators, effect modifiers, etc. when considering their models. I think the authors should consider including other a priori forced variables (gravity, for one, and perhaps others) regardless of statistical significance.

Line 272: "Variables that are missing more than 50%..." Are the authors referring to missingness in an individual study, or in the pooled dataset?

Line 273-275: Provide a reference, and describe the direction of impact/nature of the interaction.

Line 276: Is there a threshold of heterogeneity that will preclude combining data from certain sources (i.e. analogous to I-squared statistic cutoffs for aggregated meta-analyses)? If so, please specify here.

Line 283: Will the authors capture information about IPTp policy (and uptake) at each study site, and how will this be accounted for in their analyses?

Line 296: I think this can be included in the paragraph above, as a fourth sensitivity analysis.

Line 311: "...very similar" This is vague. Please specify in what way they are similar.

DISCUSSION

Line 320: Consider adding a clause to the end of the introductory sentence explaining in brief the statistical approach that makes this type of analysis possible/valid, i.e. "...by incorporating the IPD from single-arm interventional or observation studies **using [type of specialized statistical methods]**" Please include a brief 1-2 sentence discussion of the limitation of IPD meta-analyses vis a vis aggregate meta-analyses.

Line 319-329: Perhaps some of this can be moved to the Introduction (see comment above).

Line 333: Pregnant women are not a neglected population, but they are an under-studied population.

MINOR TYPOS

Line 236 – plural

Line 244 – missing "who"

Line 305 – Stata, not STATA. What version number?

Others noted throughout in addition to these above.

REVIEWER	Natalie Dean Dept of Biostatistics, University of Florida, USA
REVIEW RETURNED	31-Dec-2018

GENERAL COMMENTS	This paper describes an analysis plan for an IPD meta-analysis of malaria treatment efficacy in pregnant women. The statistical analysis is clear and adequately detailed. The success of the study will of course depend upon the willingness of other groups to share data, but it seems like there is a network in place which should facilitate this. I have only few comments: (1) The overall unadjusted log-rank test may be too heavily confounded to be meaningful, especially as there may be systematic differences in which observational study sites use which treatments. The primary effectiveness analysis would be better off as a stratified log-rank test, adjusting for study site. This then excludes any site where both treatments are not used as they cannot provide data on the treatment difference, adjusting for site. This approach would have the further benefit of “matching” the subsequent stratified analysis at fixed time points (days 28, 42, and 63). (2) The adverse events listed are generic. Is there any concern about adverse events to the fetus or adverse events that are pregnancy specific? Are these data available?
---

VERSION 1 – AUTHOR RESPONSE

Thank you for the useful comments from three reviewers. Our responses to each comment are described below. We have also modified few parts in the manuscript for clarity. Line numbers in our replies correspond to the line number of the tracked manuscript.

Reviewer: 1

Reviewer Name: A.M. van Eijk

Institution and Country: Liverpool School of Tropical Medicine, Liverpool, UK

Please state any competing interests or state ‘None declared’: None declared

Please leave your comments for the authors below

The efficacy of artemisinin-based and quinine-based treatments for uncomplicated falciparum malaria in pregnancy: a protocol for systematic review and individual patient data (IPD) meta-analysis, by Makoto Satio et al.

Overall

This is a detailed protocol of an important individual participant data analysis for malaria treatment among pregnant women. The study is relevant and examines drugs that are currently used; it uses institutionalized mechanisms to make now datasets more widely available. The paper is well written. I only have some minor questions or requests for clarification.

Authors' reply: Thank you for your thoughtful comments from your broader perspective on malaria in pregnancy. We made changes as below.

Abstract and article summary

In the abstract you mention "the lack of standardised protocols in this sub-population". In the article summary you also mention: "Limitations of this IPD meta-analysis include the potential difficulty in acquiring the IPD and the heterogeneity of the study designs, study population and parasite population. A risk of bias analysis will be conducted to address the impact of potential unshared data." I am not sure what the guidelines are for both these sections but would appreciate if some limitations could also be part of the abstract.

Authors' reply: We referred to the current WHO guidelines on assessing antimalarial efficacy, which recommend excluding pregnant women from treatment efficacy trials. There are no guidelines on how to assess treatment efficacy in pregnant women. We have added limitations in the abstract.

Introduction

".around 1500 studies.." How was this number identified?

Authors' reply: This number is based on the WWARN clinical trials publication library (<https://www.wwarn.org/tools-resources/literature-reviews/wwarn-clinical-trials-publication-library>). We have added this as a reference.

Line 84: This "Besides," can be removed, because the assessment of risk factor is one of the primary outcomes, and not a "Besides".

Authors' reply: The sentence has been revised as follows: Evidence on the treatment efficacy during pregnancy needs to be assembled.

Methods

How are observational cohort studies different from single arm studies? If all participants receive an intervention, I thought single arm studies were still interventional? Just wondering about the word observational here.

Authors' reply: When the treatment and follow-up were done following the routine local/institutional guidelines, we regarded them as observational studies, which could have more than one treatment groups. When some interventions, which were not routinely done (e.g. administering a new drug or frequent sampling in PK studies), were conducted, we regard them as single-arm interventional studies.

“P. vivax intercalated infection will be regarded as censored if the original study did not test PCR for falciparum recurrences after intercalated vivax infection. If the original study tested PCR for falciparum recurrences regardless of intercalated vivax infection, vivax infection will be regarded as a time-dependent covariate.”

This section is not clear to me. “after intercalated vivax infection” does this mean, after the P. vivax infection was detected or at the same time that the P. vivax infection was detected? If the participant received treatment for P. vivax, this may affect the treatment results for P. falciparum, so I am not sure how you can use P. vivax as a time-dependent co-variate. Would you also add treatment for P. vivax as a covariate?

Authors’ reply: We clarify the definition of P. vivax intercalated infection in the manuscript. It means P. vivax infection before the recurrence of P. falciparum. As you wrote, the treatment (and infection) for P. vivax can affect the treatment outcome of P. falciparum, therefore, the time before the P. vivax intercalated infection will be analysed separately from the time after the P. vivax intercalated infection (as a time-dependent covariate), rather than ignoring the P. vivax intercalated infection. P. vivax co-infection at baseline will be added as a covariate (Line 228).

“Likelihood ratio test (LRT) and Akaike’s Information Criterion (AIC) will be used to compare nested and non-nested models, respectively.” My apologies for possible lack in statistical background, but I would appreciate if it could be explained what the nested models are in this context, given that these are all treatment studies.

Authors’ reply: A common selection criterion for choosing between nested models (i.e. in situations where one model is a special case of another or we can say that the set of covariates of one model is a subset of the covariates of the other model) is the likelihood ratio test (LRT). A widely used alternative to the likelihood ratio test (LRT) is Akaike Information Criterion (AIC), which is equal to the log-likelihood adjusted for a number of covariates in the model.

“Interaction between gravidity (parity) and endemicity will be assessed, as the impact of gravidity (i.e. pregnancy-specific immunity) can be different depending on the endemicity.” Will age also be explored? In a previous study in an area of high malaria endemicity, we noted that low gravidity number was more important as a risk factor to get malaria, whereas young age was more important as a risk factor for treatment failure. Of course, there may be co-linearity between these two, which needs to be evaluated, but underlying mechanisms may be different, dependent on endemicity in the area.

Authors’ reply: Thank you for your suggestion. We will check the interaction between age and endemicity if age is significant in the multivariable model. This was added to the protocol as an a priori interest (Line 300-303).

Will there be an attempt to match the included studies with available molecular resistance data to artemisinins or the partner drug at the time of and in the region of the study? This may perhaps only be relevant to the Thai-Burmese border.

Authors’ reply: We will assess the impact of artemisinin resistance by referring to the prevalence of molecular marker (K-13 mutation) in the study year at the study site. We have added a sentence under the section of strength of the body of evidence.

Overall:

The variation between past and future tense is confusing throughout the manuscript, and could be more consistent. E.g. in variable selection: "Once only significant covariates remained in the model, all excluded variables will be added to this model one by one to check whether there are any variables that become significant in the presence of other risk factors." The search-section is written in the past as if this has already been conducted. So if this is part of an ongoing study, the number of potential studies identified could perhaps be added.

Authors' reply: Thank you for pointing this out. We have checked the tense. We will keep the number of potential studies for the main analysis paper, as this is one of the outcomes.

Reviewer: 2

Reviewer Name: Matthew Ippolito

Institution and Country: Johns Hopkins University, USA

Please state any competing interests or state 'None declared': None declared

Please leave your comments for the authors below

Please see attached. Interesting proposed study and I am looking forward to learning the results.

REVIEWER COMMENTS

The efficacy of artemisinin-based and quinine-based treatments for uncomplicated falciparum malaria in pregnancy: a protocol for systematic review and individual patient data (IPD) meta-analysis

BMJOPEN-2018-027503

Authors' reply: Thank you for your detailed comments. Our responses are itemised as below.

TITLE AND GENERAL COMMENTS

This is a protocol for an important and very timely review of antimalarial drugs for malaria-in-pregnancy, carried out by leading experts in the field and coming on the heel of updated WHO recommendations. The authors' approach of combining results of RCTs and observational cohort studies is

1. Efficacy suggests results of Phase II clinical trials (e.g. potency, efficacy). My personal preference would be to remove this from the title perhaps simply dropping "The efficacy of" from the title.

Authors' reply: We use the efficacy as defined by WHO (e.g. Methods and techniques for clinical trials on antimalarial drug efficacy: Genotyping to identify parasite populations). The efficacy of treatment is the primary objective of this IPD meta-analysis, so we would like to keep the original title for clarity.

2. For the authors' definition of malaria-in-pregnancy, they do not distinguish between uncomplicated malaria and asymptomatic malaria but rather group these together. They do not give sufficient attention to placental malaria, also a distinctive clinical entity. I suspect these patients groups may be different enough from each other, pathophysiologically and parasitologically, to warrant separation of the group in analyses. It may also be worthwhile to mention intermittent presumptive treatment in pregnancy (IPTp).

Authors' reply: We will differentiate the uncomplicated falciparum malaria and asymptomatic malaria infection in the analyses by including the presence of fever.

We plan to assess the direct outcome of treatment (i.e. parasitological clearance of parasitaemia) in the first place, in the similar format to what is recommended by the WHO guidelines on assessing treatment efficacy in the non-pregnant populations.

We focus on the treatment of malaria infection, rather than prevention in this pooled analysis.

3. Can the authors please include their rationale for including only studies of ACTs and quinine rather than any antimalarial agent.

Authors' reply: This is because the current WHO guidelines on treatment recommend these two treatments (quinine for the first trimester and ACTs for the second and third trimesters).

4. The authors should discuss differences between ACT and quinine pharmacokinetics and pharmacodynamics, and think about how these differences should inform their inclusions and exclusions, case definitions, treatment of missing time points, etc. Pharmacodynamics differ (e.g. parasite clearance is slower with quinine) and some ACTs have partner drugs with extremely long elimination half-lives (mefloquine, piperaquine).

Authors' reply: Because of the slow parasite clearance, early treatment failure (i.e. the presence of parasitaemia in the first seven days after the treatment) will not be applied to quinine treatment (Line 215). As this pooled analysis do not intend to analyse PK data, inclusion and exclusion criteria and case definition will not be affected by the pharmacokinetics of the drugs for assessing the efficacy, following the WHO guideline for the non-pregnant populations.

ABSTRACT

Line 30: The preferred term is "special population" when discussing pregnant women in relation to drug PK/PD (Line 30 and other locations throughout the article).

Authors' reply: We have amended accordingly.

Line 31-21: Please add safety and tolerability in pregnancy (in addition to treatment efficacy and risk factors of treatment failure).

Authors' reply: This IPD meta-analysis is primarily on the treatment efficacy. Safety is one of our secondary outcomes.

Line 36: Please spell out LILACS and other acronyms at their first instance throughout the manuscript.

Authors' reply: We have amended as requested.

Line 64: Another limitation is heterogeneity of transmission intensity among different study sites, and perhaps non-standard or missing covariates.

Authors' reply: Malaria transmission intensity will be taken into account as is described in the manuscript. Different study designs, which we mentioned in the manuscript, will result in missing information at study level, thus covers your points

Line 65: The authors' risk of bias analysis will not only evaluate the impact of potential unshared data but also critically assess quality of studies i.e. a critical assessment of each included study's methods.

Authors' reply: We have amended.

INTRODUCTION

Line 74: Please add, "Due to the lack of evidence for both efficacy and safety **of artemisinin-based combination therapy (ACT) in pregnancy** ..."

Authors' reply: We have amended

Line 77: Can the authors explicitly state, for readers' sake, that the statistical methods they propose are the preferred method for combining results of observational cohort data with data from randomized controlled trials if this is the case? In the introduction can the authors please add a paragraph explaining to readers why they have chosen an individual pooled data (IPD) meta-analysis, and briefly discuss the benefit of this approach over aggregated meta-analysis approaches that more readers are familiar with? (There is language in the Discussion that the authors can move, or adapt to, the Introduction.)

Authors' reply: The advantages and disadvantages of IPD meta-analysis are expanded in the Discussion.

Line 78: I do not think it is necessary to speculate regarding future WHO guidelines. The authors can highlight that their study will provide additional evidence for consideration. The reference (#8) is to the 2015 guidelines so I would omit this.

Authors' reply: #8 is not the 2015 WHO guidelines, but the recommendation by malaria policy advisory committee to the WHO regarding the next WHO treatment guidelines.

Line 80: "...the bulk of evidence is limited" The authors should omit this sentence, or reword to explain what evidence is lacking (safety, efficacy, dosing/PK, etc.).

Authors' reply: We have amended this sentence as described above for the first reviewer.

Line 84: Replace "Besides" with "In addition" or similar connector.

Authors' reply: We have amended.

Line 85-88: This section is not clear to me. Why are assessments of clinical efficacy in pregnant women necessarily different from clinical assessments in other populations? WHO defines adequate clinical and parasitological response, early treatment failure, late clinical failure, and late parasitological failure, etc. Is there really a lack of standard methodology? Perhaps if the authors discuss the delineation between uncomplicated malaria, asymptomatic malaria, and placental malaria it will resolve this sticking point.

Authors' reply: The WHO guidelines on assessment of treatment efficacy specifies that pregnant women should be excluded. This IPD meta-analysis is aiming to provide the summary of the treatment efficacy in pregnancy following the WHO guidelines for the non-pregnant populations, given the differences of the adopted study designs that each study used.

OBJECTIVES

Line 107-109: Since they are important secondary outcomes, perhaps the authors should review the literature and discuss gametocyte carriage in pregnant women, and risk of vivax malaria following treatment of falciparum malaria, in their Introduction

Authors' reply: Our primary objective is to summarise the treatment efficacy. We have revised it for clarity.

Line 108: Suggest, changing "risk of adverse events..." to "safety and tolerability of artemisin- and quinine-based treatments"

Authors' reply: We have amended accordingly.

METHODS AND ANALYSES

Line 113: The authors may want to consider including studies even with shorter followup times than 28 d because there may be safety and efficacy outcomes, gametocyte outcomes etc. reported in those studies.

Authors' reply: We followed the current WHO guidelines to assess the treatment efficacy, which requires minimum 28 days of follow-up. Any shorter follow up will likely miss a substantial amount of recurrence and will overestimate the true efficacy of the drug tested.

Line 116: The authors should pre-specify which genotyping approaches they will accept (e.g. all three of mps1, mps2, glurp).

Authors' reply: We will not explore the technical difference between different PCR methodologies in this pooled analysis.

Line 118: Can the authors justify why they are excluding studies with ≤ 10 eligible

pregnant women? Is this related to assumptions about baseline hazard/shared frailty approach? If this is an arbitrary number, can the authors remove this exclusion?

Authors' reply: According to the PRISMA-IPD guidelines, exclusion of small studies is justified considering the very little contribution to the whole estimate and considerable effort to obtain the data.

Line 119: What about displaced or migrant populations? These populations are commonly excluded because of differences in premunition, etc.

Authors' reply: This is a secondary use of data and considers only studies which have received ethical approval to be conducted. Exclusion criteria apply to answer the objectives of the study.

Line 122: Is there a cutoff for parasitemia (e.g. >200,000 or >4% meets definition of severe). What about mono vs. mixed Plasmodium infections?

Authors' reply: Both baseline parasitaemia and mixed infection will be assessed (Line 220-).

Line 123: The authors must think more carefully about this. Asymptomatic malaria is distinct from uncomplicated malaria and strong justification is needed if the authors intend to combine these two patient groups. They may want to exclude asymptomatic malaria or stratify by uncomplicated vs. asymptomatic malaria.

Authors' reply: Presence of fever will be assessed as a covariate.

Line 126: The authors forgot to include gametocyte-related outcomes. They may want to add hemoglobin recovery, QT prolongation. They could expand adverse events to include pregnancy complications, adverse events that affect the infant or fetus. They may want to include pharmacokinetic and pharmacodynamic data if sufficient studies with these measures are identified.

Authors' reply: This IPD meta-analysis focuses on the treatment efficacy in the first place. Analysis of other important outcomes (e.g. PK, haematological change, cardiotoxicity) is out of scope.

Line 130-142: The search be conducted within 12 months of the intended publication date of the meta-analysis itself. Can the authors update all of their searches (Scopus, LILACS, etc.) since these were last done over 2 years ago in July 2016. Please add the date range for the search including beginning date i.e. was it all studies since inception of each database?

Authors' reply: We have conducted the search without limitation on the language and publication year. We will conduct a final literature search before the publication.

Line 142: May want to add, "... based on studies identified by the final updated search **for which patient-level data are made available**"

Authors' reply: We removed this sentence.

Line 149: Can the authors please very briefly state, in 1-2 sentences, what these terms and conditions are? It can help readers understand why a certain PI may or may not have agreed to contribute data.

Authors' reply: We have added the information.

Line 151: It is just a minor suggestion but the authors may also want to include that the WWARN includes a mechanism for contributors and other researchers to request access to the WWARN data.

Authors' reply: The WWARN data access policy is available here (<https://www.wwarn.org/tools-resources/terms-data-access>). This is not relevant to this protocol, so we will not add this information in this protocol.

Line 155: Minor typo, "REsolved"

Authors' reply: Thank you for pointing this out. We have corrected it.

Line 161: Add additional bullets here for completeness e.g. gravidity/parity, directly observed therapy or not, recent antimalarial drugs (should this be an exclusion criterion?) including whether IPTp was administered or not, geographic region including information about transmission intensity, mono vs. mixed Plasmodium infection, parasite density, hemoglobin concentration, type of malaria (uncomplicated, asymptomatic, placental—see comments above), parasite clearance.

Authors' reply: These will be assessed as covariates, as listed under the section "Variables and their definitions".

Line 172: Why are the authors keeping those with hyperparasitemia? These cases should be clinically regarded as severe malaria, even if they were not treated as such in a particular study. The authors could consider a subgroup analysis, or sensitivity analysis excluding these hyperparasitemic patients, if the authors retain them in their dataset.

Authors' reply: Patients with hyperparasitaemia will be assessed if they were included in the original studies. We will use a binary covariate to distinguish them from non-hyperparasitaemic patients and will explore if treatment effects are different in this group.

Line 174: How will the researchers define "incomplete dose" and should the authors consider including those patients? This could become an important source of bias e.g. incomplete doses might be due to adverse event/intolerability of the medication, disease progression requiring change in therapy, or other events that the authors would want to capture.

Authors' reply: We will consider conducting separate analyses including only those who got complete dosing and including all patients, when there are significant amount of patients who did not complete the dose. This has been added in Line 231-

Line 176-179: I believe the primary outcome should be stated as PCR corrected clinical outcome (ACPR, ETF, LCF, LPF). Do other secondary outcomes include adverse events, hemoglobin recovery, pregnancy outcome, placental malaria, infant/fetus AEs, etc.? This appears to be the first mention of parasite clearance (Line 178).

Authors' reply: We clarified that our primary outcome is PCR-corrected treatment efficacy. We believe parasite clearance is one of the measurements of treatment efficacy mentioned as the primary objective.

Line 178: Why is uncorrected Pf recurrence included here? The authors stated that they are only including studies for which genotyping data are included for PCR correction of clinical outcomes (i.e. to distinguish reinfection vs. recrudescence).

Authors' reply: This is recommended by the WHO, therefore we put this as a secondary outcome.

Line 181: The authors should consider competing risk analysis rather than censoring to deal with reinfections +/- indeterminate PCRs. Censoring those with indeterminate PCRs is going to potentially introduce bias, and note of this should be made under limitations.

Authors' reply: We understand the methodological limitations. However, we are simply following the current WHO guidelines, which recommends to censor reinfection.

Recent works by our colleague, Prabin Dahal (submitted for publication) demonstrated that results are very similar between conventional analysis (as we propose) and competing event approach in case of treatment failure rate <10% & low/moderate transmission intensity settings. These results are supported by a direct comparison of the results generated using competing risk analysis to that from a conventional survival analysis for 92 published studies, and also through a simulation study. We provide the details of these upcoming publications as future reference:

1. WWARN Methods Study Group. Competing risk events in antimalarial studies of uncomplicated *Plasmodium falciparum* malaria.
2. Prabin Dahal, Philippe J. Guerin, Ric N. Price, Julie A. Simpson, Kasia Stepniewska. Evaluating antimalarial efficacy in single-armed and comparative drug trials using competing risk survival analysis: A simulation study.

These two forthcoming publications suggested that the derived estimates of drug failure were different only up to two decimal places in the scenario which is relevant for our study, which is considered as being clinically less meaningful. This scenario corresponds to the data we expect to be contributing to this IPD meta-analysis.

Line 188: I do not think it is safe to assume a missing weekly smear between two negative weekly smears is also negative unless the authors are performing repeated failure/event survival analysis in which case a participant can enter and exit the risk set at different time points. (They propose repeated episodes analysis later in the manuscript, Line 234, but could be explicitly mention in the Introduction or other, earlier section.) However, I understand their approach to define any LPF as a failure event; and since it is not safe to assume that a missing smear = negative even if couched between two negative smears, then those participants should probably be censored at the last available observation before the missing observation.

Authors' reply: This approach is supported by biological ground. Considering the cycle of *plasmodium falciparum* in human, it is unlikely that a patient with a negative slide on week 2 and week 4 would have presented with a positive parasitemia on week 3 without any symptom and treatment. This approach has been validated by a group of experts while establishing the WWARN data management plan for curation, has been repeatedly published and accepted by the malaria research community as a reasonable estimate.

Line 189-191: Why 31 days? The authors should choose a range based on parasite biology and drug pharmacodynamics (e.g. some ACT partner drugs, and quinine, have very long half-lives).

Authors' reply: This is a pragmatic decision which allows us to include studies with monthly follow up (28+/- 3 days).

Line 196: Correct “Early late treatment failure” to “Early treatment failure”

Authors’ reply: We have clarified this as follows: the presence of parasitaemia within the first 7 days.

Line 198: The authors should capture all reported AEs rather than a list of prespecified AEs, or otherwise justify why they are using a proscribed list. It would also be interesting to discuss post-artemisinin hemolysis which can also occur with oral agents.

Authors’ reply: As we are pooling data from different studies, and considering the heterogeneity of AEs reporting, only a pre-defined sets of AEs can be standardised and assessed. Haematological change will not be assessed in this pooled analysis.

Line 206: What about gametocytes that emerge after start of treatment in those who were initially without gametocytemia?

Authors’ reply: This will be assessed as a secondary outcome as we specified.

Line 208: Are the authors capturing information on recent antimalarial treatment within the past week or 2 weeks, etc.?

Authors’ reply: Yes, we are aiming to capture the history of antimalarial use, either at study level or individual level, though it is important to note that this is poorly reported in primary studies. The manuscript has been amended.

Line 216: The authors may want to consider *P. vivax* infection as a competing risk, rather than censoring (see comment above).

Authors’ reply: We will be following the current WHO guideline as above.

Line 224: Range, or interquartile range? What about geometric mean for parasite counts (preferred by many in the field), or PK data?

Authors’ reply: We have specified it. Skewed variables including parasitaemia will be (log-)transformed or categorised in the analyses.

Line 228: My instinct is to drop the analysis at fixed time points (28, 42, 63 d).

Authors’ reply: This is for the descriptive purpose and those time points are usually reported in literature. Combining the comments from two reviewers (reviewer 2 and 3), we have dropped the comparison of Kaplan Meier curves (by log rank test).

Results

can be misleading—the authors are going through the rigor of collecting longitudinal data so why reduce it to cross-sectional data even in an exploratory analysis? I would favour dropping this altogether.

Authors' reply: There are only few studies with longitudinal recording of malaria episodes. The majority of the studies have a single study episode for each woman. This is not cross-sectional data, but data with at least 28 days of follow-up.

Line 250: For mixed effects, can the authors state which are fixed and which are random?

Authors' reply: Only study sites will be fitted as random effects.

Line 254: Adverse events, not adverse symptoms.

Authors' reply: We have amended.

Line 259: "Symptoms developed at any time during the study period may be added."

Please change to "Signs or symptoms", and please explain when they will or will not be added, or omit this sentence altogether.

Authors' reply: The sentence has been omitted.

Line 262: This approach is falling out of favor (referring to the authors' proposed strategy for selecting covariates). The authors refer to "risk factors" (predictors) but ignore potential confounders, mediators, effect modifiers, etc. when considering their models. I think the authors should consider including other a priori forced variables (gravity, for one, and perhaps others) regardless of statistical significance.

Authors' reply: We will include baseline parasitaemia and treatment as a priori forced covariates. Interaction of gravity and endemicity is also our a priori interest as specified in the manuscript.

Line 272: "Variables that are missing more than 50%..." Are the authors referring to missingness in an individual study, or in the pooled dataset?

Authors' reply: In the pooled dataset.

Line 273-275: Provide a reference, and describe the direction of impact/nature of the interaction.

Authors' reply: A reference is added.

Line 276: Is there a threshold of heterogeneity that will preclude combining data from certain sources (i.e. analogous to I-squared statistic cutoffs for aggregated metaanalyses)? If so, please specify here.

Authors' reply: There are no guidelines on this for one-stage IPD meta-analysis, particularly as we are using shared frailty model. We will provide these assessment results as a reference.

Line 283: Will the authors capture information about IPTp policy (and uptake) at each study site, and how will this be accounted for in their analyses?

Authors' reply: We will capture IPTp information at least at study level, and possibly at individual level if available. Any antimalarial use including IPTp-SP will be censored on the day in the analysis following the WHO guidelines.

Line 296: I think this can be included in the paragraph above, as a fourth sensitivity analysis.

Authors' reply: We have amended the sentence.

Line 311: "...very similar" This is vague. Please specify in what way they are similar.

Authors' reply: We have amended.

DISCUSSION

Line 320: Consider adding a clause to the end of the introductory sentence explaining in brief the statistical approach that makes this type of analysis possible/valid, i.e. "...by incorporating the IPD from single-arm interventional or observation studies **using [type

of specialized statistical methods]**" Please include a brief 1-2 sentence discussion of the limitation of IPD meta-analyses vis a vis aggregate meta-analyses.

Authors' reply: The limitations of IPD meta-analyses are mentioned in Lines 365 (increased time and effort of gathering and standardising the IPD).

Line 319-329: Perhaps some of this can be moved to the Introduction (see comment above).

Authors' reply: We would like to keep the description of methodological advantages and limitations of IPD in the discussion.

Line 333: Pregnant women are not a neglected population, but they are an understudied population.

Authors' reply: Thank you for this important point. We have revised them accordingly.

MINOR TYPOS

Line 236 – plural

Line 244 – missing "who"

Line 305 – Stata, not STATA. What version number?

Authors' reply: Thank you for pointing them out. We have corrected them.

Others noted throughout in addition to these above.

Reviewer: 3

Reviewer Name: Natalie Dean

Institution and Country: Dept of Biostatistics, University of Florida, USA

Please state any competing interests or state 'None declared': None declared

Please leave your comments for the authors below

This paper describes an analysis plan for an IPD meta-analysis of malaria treatment efficacy in pregnant women. The statistical analysis is clear and adequately detailed. The success of the study will of course depend upon the willingness of other groups to share data, but it seems like there is a network in place which should facilitate this. I have only few comments:

Authors' reply: Thank you for your comments from your statistical point of view, which are helpful for us to refine our statistical plan.

(1) The overall unadjusted log-rank test may be too heavily confounded to be meaningful, especially as there may be systematic differences in which observational study sites use which treatments. The primary effectiveness analysis would be better off as a stratified log-rank test, adjusting for study site. This then excludes any site where both treatments are not used as they cannot provide data on the treatment difference, adjusting for site. This approach would have the further benefit of "matching" the subsequent stratified analysis at fixed time points (days 28, 42, and 63).

Authors' reply: We agree that crude Kaplan-Meier curves and log-rank test will be misleading. We have omitted these sentences, and we will present the pooled K-M estimates for each treatment at fixed time points stratified by study sites.

(2) The adverse events listed are generic. Is there any concern about adverse events to the fetus or adverse events that are pregnancy specific? Are these data available?

Authors' reply: Pregnancy outcomes may be assessed if enough data are available (if enough studies collected pregnancy outcomes and necessary relevant information such as gestational age at delivery, birthweight or sex). Otherwise, we will report crude prevalence.

VERSION 2 – REVIEW

REVIEWER	AM van Eijk Liverpool School of Tropical Medicine, UK
REVIEW RETURNED	08-Apr-2019

GENERAL COMMENTS	My questions have been answered satisfactorily. I have no new comments. The manuscript has improved and is clear.
---

REVIEWER	Matthew Ippolito Johns Hopkins, USA
REVIEW RETURNED	30-Apr-2019

GENERAL COMMENTS	The authors sufficiently addressed reviewers' concerns.
---

REVIEWER	Natalie Dean Department of Biostatistics, University of Florida, USA
REVIEW RETURNED	16-Apr-2019

GENERAL COMMENTS	This will be a useful protocol that will provide informative data on malaria treatment for pregnant women. The paper is clearly written and thorough. The authors have satisfactorily responded to my earlier comments, and I have no new comments to add.
--